# Status and Distribution of Diseases Caused by Phytoplasmas in Africa

**DOI:** 10.3390/microorganisms13061229

**Published:** 2025-05-27

**Authors:** Shakiru Adewale Kazeem, Agnieszka Zwolińska, Joseph Mulema, Akindele Oluwole Ogunfunmilayo, Shina Salihu, Joy Oluchi Nwogwugwu, Inusa Jacob Ajene, Justina Folasayo Ogunsola, Adedapo Olutola Adediji, Olubusola Fehintola Oduwaye, Kouamé Daniel Kra, Mustafa Ojonuba Jibrin, Wei Wei

**Affiliations:** 1Post-Entry Quarantine, Diagnostic and Surveillance Station, Nigeria Agricultural Quarantine Service, Ibadan 200273, Oyo State, Nigeria; 2Department of Plant Physiology, Faculty of Biology, Adam Mickiewicz University, 61-712 Poznan, Poland; 3CABI, Canary Bird, 673 Limuru Road, Muthaiga, Nairobi 633-00621, Kenya; 4National Cereal Research Institute, Ibadan Research Station, Ibadan 200273, Oyo State, Nigeria; 5Department of Forest Conservation and Protection, Forestry Research Institute of Nigeria, Ibadan 200272, Oyo State, Nigeria; 6International Centre of Insect Physiology and Ecology, Nairobi 00100, Kenya; 7Department of Biological Sciences, Bells University of Technology, Ota 112104, Ogun State, Nigeria; 8Research Office, Pan African University Life and Earth Sciences (Including Health and Agriculture), Ibadan 200284, Oyo State, Nigeria; adedapo.adediji@gmail.com; 9Institute of Agricultural Research and Training, Obafemi Awolowo University, Moor Plantation, Ibadan 200273, Oyo State, Nigeria; 10Natural Sciences Training and Research Unit, Plant Health Laboratory, University Nangui Abrogoua, 02 BP 801 Abidjan, Côte d’Ivoire; 11Department of Entomology and Plant Pathology, Oklahoma State University, 127 Noble Research Center, Stillwater, OK 74084, USA; 12Molecular Plant Pathology, USDA, Agricultural Research Service, Beltsville, MD 20705, USA; wei.wei@usda.gov

**Keywords:** “*Candidatus* Phytoplasma”, 16S rRNA RFLP, Africa, lethal yellow

## Abstract

Phytoplasma (“*Candidatus* Phytoplasma” species) diseases have been reported globally to severely limit the productivity of a wide range of economically important crops and wild plants causing different yellows-type diseases. With new molecular detection techniques, several unknown and known diseases with uncertain etiologies or attributed to other pathogens have been identified as being caused by Phytoplasmas. In Africa, Phytoplasmas have been reported in association with diseases in a broad range of host plant species. However, the few reports of Phytoplasma occurrence in Africa have not been collated together to determine the status in different countries of the continent. Thus, this paper discusses the geographical distribution, detection techniques, insect vectors, alternative hosts and socio-economic impacts of Phytoplasma diseases in Africa. This is to create research perspectives on the disease’s etiology in Africa for further studies towards identifying and limiting their negative effects on the continent’s agricultural economy. In Africa, Phytoplasmas recorded in different countries affecting different crops belong to eight groups (16SrI, 16SrII, 16SrIII, 16SrIV, 16SrVI, 16SrXI, 16SrXIV and 16SrXXII) out of the 37 groups and over 150 subgroups reported worldwide on the basis of their 16S rRNA RFLP profile. Lethal yellow disease was the most destructive Phytoplasma reported in Africa and has a high socio-economic impact.

## 1. Overview and Rationale

Phytoplasma (“*Candidatus* Phytoplasma” species) diseases have been reported globally to severely limit the productivity of a wide range of economically important crops and wild plants, causing different yellows-type diseases [1,2,3,4,5,6,7,8,9,10]. Table 1 lists the Phytoplasma groups reported worldwide on the basis of their 16S rRNA genes. New Phytoplasma-associated diseases are continuously being discovered, mainly because of improved molecular diagnostic methods. With these detection techniques, several unknown and known diseases with uncertain etiologies or attributed to other pathogens have been identified as being caused by Phytoplasmas. Many plant species affected by yellows-type diseases were wrongly attributed to viruses because of their infective spread, symptoms and transmission by insects [11,12].

Doi et al. (1967) reported that the etiological agents of yellow diseases could be wall-less prokaryotes related to bacteria rather than viruses [13]. These organisms were first referred to as mycoplasma-like organisms (MLOs) because of their similarity to mycoplasmas infecting animals, which also belong to the Mollicutes class [3,14]. In contrast to mycoplasmas, MLOs were nutritionally fastidious and phylogenetically related to the Gram-positive bacteria [15,16]. The MLOs were eventually named “*Candidatus* Phytoplasma” after the interpretation of different molecular data [2]. These bacteria belong to the super kingdom Prokaryota; kingdom Monera; domain Bacteria; phylum Firmicutes (low G+C, Gram-positive eubacteria); class Mollicutes and Candidatus (Ca.) genus Phytoplasma [17].

Phytoplasmas are wall-less prokaryotes, insect-vectored and phloem-limited bacteria with A+T-rich genomes that are 530–1350 kb in size [4,16,18,19,20,21]. These bacteria differ from Gram-negative insect-vectored proteobacteria such as liberibacters, phlomobacters [9,22] and Spiroplasmas, which are culturable in vitro [9,23].

**Table 1 microorganisms-13-01229-t001:** Phytoplasma strains reported worldwide.

Phytoplasma Strain (Host Plant)	16Sr	Related Ca. Species	Origin
Blackcurrant reversion (*Ribes nigrum*)	I-C	*Phytoplasma asteris*	Czech.
Clover phyllody-England (*Trifolium* sp)	I-B	*Phytoplasma asteris*	UK
Apricot chlorotic leaf roll (*Prunus armeniaca* L.)	I-F	*Phytoplasma asteris*	Spain
Atypical aster yellows (various plants)	I-M	*Phytoplasma asteris*	Germany
Lime witches’-broom (*Citrus aurantifolia*)	II-B	*Phytoplasma aurantifolia*	Arabia
Faba bean phyllody (*Vicia faba* L.)	II-C	*Phytoplasma aurantifolia*	Sudan
Crotalaria saltiana phyllody (*Crotalaria saltiana*)	II-C	*Phytoplasma aurantifolia*	Sudan
Soybean phyllody (*Glycine max* (L.) Merr.)	II-C	*Phytoplasma aurantifolia*	Thailand
Australian tomato big bud (*Solanum lycopersicum*)	II-D	*Phytoplasma aurantifolia*	Australia
Sweet potato little leaf (*Ipomoea batatas*)	II-D	*Phytoplasma aurantifolia*	Australia
Ipomoea (unspecified)	II-D	*Phytoplasma aurantifolia*	Fiji
Peach western X (*Prunus persica*)	III-A	*Phytoplasma pruni*	USA
Green valley X (most Stone fruits)	III-A	*Phytoplasma pruni*	USA
Poinsettia branching factor (*Euphorbia pulcherrima*)	III-H	*Phytoplasma pruni*	USA
Coconut lethal yellowing (*Adonidia merrillii*)	IV-A	*Phytoplasma palmae*	Serbia
Coconut lethal yellowing (*Hyophorbe verschafeltii*)	IV-A	*Phytoplasma palmae*	USA
Coconut lethal yellowing (*Phoenix rupicola*)	IV-A	*Phytoplasma palmae*	USA
Tanzanian lethal decline (*Cocos nucifera* L.)	IV-B	*Phytoplasma cocostanzaniae*	Tanzania
Ghanaian Cape St. Paul wilt (*Cocos nucifera* L.)	IV-C	*Phytoplasma cocosnigeriae*	Ghana
Elm witches’-broom (*Ulmus* sp.)	V-A	*Phytoplasma ulmi*	France
Potato witches’-broom (*Solanum*)	VI-A	*Phytoplasma trifolii*	USA
Brinjal little leaf	VI-A	*Phytoplasma trifolii*	India
Catharanthus phyllody (*Catharanthus roseus*)	VI-C	*Phytoplasma trifolii*	Sudan
Ash yellows (*Fraxinus* sp.)	VII-A	*Phytoplasma fraxini*	USA
Loofah witches’ broom (*Luffa aegyptica* Mill.)	VIII-A	*Phytoplasma luffae*	Australia
Pigeon pea witches’-broom (*Cajanus cajan*)	IX-D	*Phytoplasma phoenicium*	-
Apple proliferation (*Malus domestica*)	X-A	*Phytoplasma mali*	USA
German stone fruit yellows (*Prunus* sp.)	X-B	*Phytoplasma prunorum*	Italy
European stone fruit yellows (*Prunus persica*)	X-B	*Phytoplasma prunorum*	Germany
Napier grass stunt (*Pennisetum purpureum*)	XI	*Phytoplasma oryzae*	Germany
Cordyline Phytoplasma (*Fragaria ananassa*)	XII	*Phytoplasma fragariae*	Ethiopia
Stolbur of pepper (*Capsicum annuum*)	XII-A	*Phytoplasma solani*	Jersey
Mexican periwinkle virescence (*Catharanthus roseus*)	XIII	*Phytoplasma hispanicum*	Mexico
Bermuda grass white leaf (*Cynodon dictylon*)	XIV-A	*Phytoplasma cynodontis*	-
Hibiscus witches’ broom (*Hibiscus rosa-sinensis*)	XV-A	*Phytoplasma brasiliense*	-
Sugarcane yellow leaf syndrome (*Saccharum officinarum*)	XVI-A	*Phytoplasma graminis*	-
Papaya bunchy top (*Carica papaya*)	XVII-A	*Phytoplasma caricae*	-
American potato purple top wilt (*Solanum tuberosum*)	XVIII-A	*Phytoplasma americanum*	-
Japanese chestnut witches (*Castanea crenata*)	XIX-A	*Phytoplasma castaneae*	-
Buckthorn witches’ broom (*Rhamnus catharticus*)	XX-A	*Phytoplasma rhamni*	-
Pineshoot proliferation (*Pinus halepensis*)	XXI-A	*Phytoplasma pini*	-
Nigerian coconut lethal decline (*Cocos nucifera* L.)	XXII-A	*Phytoplasma palmicola*	-
Buckland Valley grapevine yellows (*Vitis vinifera* L.)	XXIII-A	*Unnamed*	-
Sorghum bunchy shoot (*Sorghum bicolor* (L.) Moench)	XXIV-A	*Unnamed*	-
Weeping tea tree witches’ (*Leptospermum brachyandrum*)	XXV-A	*Unnamed*	-
Mauritius sugarcane yellows D3T1(*Saccharum officinarum* L.)	XXVI-A	*Unnamed*	-
Mauritius sugarcane yellows D3T2(*Saccharum officinarum* L.)	XXVII-A	*Unnamed*	-
Havana derbid	XXVIII-A	*Unnamed*	-
Cassia witches’ broom (*Cassia italica*)	XXIX-A	*Phytoplasma omanense*	-
Salt cedar witches’ broom (*Tamarix chinensis* Lour)	XXX-A	*Phytoplasma tamaricis*	-
Soybean stunt phytoplasma (*Glycine max*)	XXXI-A	*Phytoplasma costaricanum*	-
Malaysian periwinkle virescence (*Catharanthus roseus*)	XXXII-A	*Phytoplasma malaysianum*	-
Allocasuarina (*Allocasuarina mulleriana*)	XXXIII-A	*Phytoplasma allocasuarinae*	-
Grapevine yellows	XXXIV	*Abolished*	-
Pepper witches’-broom	XXXV	*Abolished*	-
foxtail palm yellow decline (*Wodyetia bifurcata*)	XXXVI	*Phytoplasma wodyetiae*	-
Stylosanthes little leaf (*Solanum tuberosum* L.)	XXXVII	*Phytoplasma stylosanthis*	-
Bogia coconut syndrome (*Cocos nucifera*)	XXXVIII	*Phytoplasma noviguineense*	-
Palm lethal wilt (*Dypsis poivreana*)	XXXIX	*Phytoplasma dypsidis*	-

- = unavailable; Source: [24,25,26].

The detection of Phytoplasmas in diseased plants was previously based on electron microscope observations, symptom expressions and transmission via insects, graft or dodder but currently uses molecular tools [23,27,28,29]. The use of molecular detection techniques has provided not only a basis for the identification of Phytoplasmas but also a reliable tool for their differentiation and classification. This technique, which is based on restriction fragment length polymorphism (RFLP) analysis of the polymerase chain reaction (PCR) target gene sequence, in silico RFLP and the online *i*PhyClassifier and CpnClassiPhyR tools have been used to classify Phytoplasmas that cause various diseases into groups and subgroups [30,31,32,33]. Recently, multilocus sequence analyses have been used to study the population structure of Phytoplasmas [24,34,35,36]. Compared with those of thousands of known Phytoplasma strains, only few strains have their genomes sequenced and documented [24,37].

Phytoplasmas have been reported to cause diseases in different plant species across many African countries [29,38]. The first recorded occurrence was in 1917, known as Bronze leaf wilt (now called Awka wilt disease), which causes lethal yellowing-type disease in coconut [38]. Subsequently, coconut lethal yellowing was similarly reported in different African countries [39]. This paper will attempt to identify and summarize the different status of reported Phytoplasma diseases in Africa. Hopefully, this will help to highlight the current status of plant pathogenic Phytoplasmas in Africa and identify gaps for future research.

## 2. Symptoms and Spread of Phytoplasma Diseases

### 2.1. Symptoms of Infection

Phytoplasmas cause virus-like symptoms in plants, and for many years, these diseases have been attributed to viruses [12]. Symptom expression of Phytoplasma infection alone cannot, in most cases, be used for definite identification of the causal organism. This is because symptom expression depends on the host species, growth stage of the host infection and strain of the Phytoplasma. Additionally, dual or mixed infections involving related or unrelated Phytoplasmas are known to occur naturally in plants [16]. Therefore, molecular diagnosis is necessary to confirm the causal Phytoplasma species since consistent isolation in axenic media is yet to be established for most Phytoplasmas [16,40,41,42,43].

Phytoplasma-infected plants present symptoms that point to severe disruptions in the usual equilibrium of growth regulators or plant hormones [3,44,45]. The symptoms induced in diseased plants vary with the Phytoplasma and with the stage of infection. Some plant species are tolerant or resistant to Phytoplasma infections with no or mild symptoms [45]. The protocol by Ermacora and Osler [45] provides descriptions and pictorial representatives of each symptom, factors influencing Phytoplasma symptom expression and practical procedures for the diagnosis of each symptom. Wei et al. [46] developed a web-based link called the Phytoplasma disease and symptom database (iPhyDSDB) to provide images and descriptive definitions of symptoms caused by Phytoplasma as a reference point to match Phytoplasma symptoms to aid virtual diagnosis. The most common and representative indicators of Phytoplasma infection include yellowing of the plants, stunting (small flowers and leaves and shortened internodes), witches’ broom (bunchy growth at stem apices due to loss of apical dominance or proliferation of auxiliary or axillary shoots/buds), phyllody (development of floral parts into green leaf-like tissues) and virescence (greening of flowers due to loss of normal flower pigments) [46]. Multiple symptoms can sometimes occur on the same host due to single infection or coinfection by multiple Phytoplasma species, as reported, for example, in wild grasses in East Africa [27]. Symptoms also include leaf curling, crinkling or cupping upwards or downwards, purple top (reddening of leaves and stems), phloem necrosis, dieback, sterility of flowers and abnormal internode elongation [16,44,45,47,48].

### 2.2. Phytoplasma Transmission

Phytoplasmas can be introduced into new geographic regions by long-distance dispersal and spread within/between fields via insect vectors, infected planting material and transovarial transmission [49,50].

#### 2.2.1. Insect Vectors

Insect vectors are the main carriers and distributors of Phytoplasmas in nature and within fields. The geographical distribution and host range of Phytoplasmas are strongly dependent upon the insect vectors found in that area and whether the insects are monophagous, oligophagous or polyphagous in their feeding habits [14,16]. Monophagous and oligophagous insect species are generally more efficient vectors than polyphagous insect species are [51]. Vectors can be found on leaves, flowers, fruits, bark, and sometimes underground in the roots of host plants [49].

Phytoplasma vectors belong to the order Hemiptera and are transmitted mainly by leafhoppers (Auchenorrhyncha: Cicadellidae) and less commonly by planthoppers (Auchenorrhyncha: Fulgoromorpha) and psyllids (Sternorrhyncha: Psyllidae), which feed on the phloem sap of infected plants [16,49,51,52,53,54,55]. Adults and nymphs can transmit Phytoplasmas since both have similar feeding behaviors [55] in a circulative-propagative manner that involves a latent period from 2 to 8 weeks [16,56]. The mere detection of Phytoplasmas in an insect does not imply that the insect is a vector; a transmission assay is needed to provide conclusive evidence [51]. This depends on insect vector competence (the ability to acquire and transmit Phytoplasmas by overcoming the insect gut and salivary gland cell barriers to becoming infectious) [51,57].

Phytoplasmas are usually transmitted by specific insect vector species, but some can transmit more than one type of Phytoplasma to the same or a range of plant species in different regions [14,29,43,55]. The vector competence of more than half of the confirmed Phytoplasma groups has not been determined, which has limited the identification of vectors of many Phytoplasmas [43,51]. This has also affected the study of insect vector ecology and the epidemiology of plant diseases caused by Phytoplasmas. Surveying vectors to determine associated Phytoplasma diseases in a given region is important for quarantine/management purposes.

The screening of insect species to confirm their vector competence is usually performed via a variety of methods [43,49]. The choice of method is determined by the insect taxon, live stage of concern, and purpose of the study. Two or three methods have to be used together if there is little prior knowledge concerning the insect vector(s) [49]. Laboratory techniques for screening vectors that can be adopted to suit less equipped laboratory environments are described by Kingdom [58], Bosco and Tedeschi [59], Bertin and Bosco [60], Kruger and Fiore [49] and Pagliari et al. [61].

Methods of capturing and storing the insect vectors of Phytoplasmas and criteria for choosing techniques were described by Weintraub and Jürgen [62] and Kruger and Fiore [49]. General and specific methods for raising insect vector colonies and maintaining Phytoplasmas were highlighted by Kingdom [58]. Both Bosco and Tedeschi [59] and Pagliari et al. [61] described vector rearing techniques and transmission experiments using insects from Phytoplasma-free laboratory colonies or field collections. The identification of all stages of insect vector species involved in Phytoplasma transmission was demonstrated by Bertin and Bosco [60] via molecular identification tools where a morphological taxonomic expert was not available.

Insect vectors transmitting Grapevine yellows disease in Tunisia and South Africa [63,64] and Napier grass stunt in Kenya and Ethiopia [65,66,67] have been confirmed (Table 2). Those suspected to transmit the lethal yellowing disease of coconut in Mozambique, Ghana and Tanzania were reported by Philippe et al. [68], Bila et al. [69] and Gurr et al. [28], respectively. The insect vector(s) for many of the reported Phytoplasmas in Africa are yet to be identified, mainly because the vectors are often not studied. Thus, identifying these vectors and possibly unknown Phytoplasmas for screening potential insect vectors were advocated by Trivellone and Dietrich [70] and Trivellone et al. [71]. This offers the opportunity to unravel Phytoplasma species and their hosts in Africa through the monitoring and diagnosis of insect vector(s) as a target for management.

**Table 2 microorganisms-13-01229-t002:** Insect vectors known to transmit Phytoplasma disease in Africa.

Insect Vector	Phytoplasma Disease	References
*Mgenia fuscovaria*, *Aconurella prolixa*	Grapevine yellows disease	[63,72,73]
*Maiestas banda* *Leptodel phaxdymas* *Exiti anus*	Napier grass stunt Phytoplasma	[65,66,67]
*Hebata decipiens*	Goosegrass white leaf	[74]

#### 2.2.2. Other Modes of Transmission

Phytoplasmas may also be transmitted from infected plants to healthy plants through the parasitic plant dodder (*Cuscuta* sp.) and grafting [75,76,77,78]. It can be spread by vegetative propagation through cuttings, tubers, rhizomes, bulbs, etc. It cannot be transmitted mechanically by inoculation with Phytoplasma-containing sap.

Transovarial and seed transmission of Phytoplasmas has been demonstrated [50,79] to be possible in the introduction and spread of Phytoplasma diseases worldwide [36]. Thus, the National Plant Protection Organization of each country should now consider these modes of Phytoplasma transmission in their pest risk analysis (the process of evaluating biological or other scientific and economic evidence to determine whether an organism is a pest, whether it should be regulated, and the strength of any phytosanitary measures to be taken against it (ISPM 5)) for safe movement of plants across international and national borders.

The possibility that Phytoplasmas are transmitted in seeds has also been demonstrated in several crops, such as coconut embryos, alfalfa (*Medicago sativa*), lime (*Citrus aurantifolia*), tomato (*Lycopersicum esculentum*), corn (*Zea mays*) and pea (*Pisum sativum*) seeds [79,80,81,82,83]. Some of these seeds contain Phytoplasmas belonging to the ribosomal groups 16SrI, 16SrXII and 16SrII [79,80,82,83,84,85].

Transovarial transmission of Phytoplasma in eggs, newly hatched nymphs and adults has been demonstrated in some insect vector/plant host combinations [50]. For example, *Scaphoideus titanus* was shown to transmit Phytoplasma transovarially to *Vicia faba* seedlings. In addition, *Hishimonoides sellatiformis* and *Matsumuratettix hiroglyphicus* transovarially vectored mulberry dwarf Phytoplasmas and sugarcane white leaf disease, respectively [50,86]. Phytoplasma prunorum and Phytoplasma mali, transmitted by *Cacopsylla pruni* and *Cacopsylla picta*, respectively, were also shown to have this type of Phytoplasma transmission [50,87].

## 3. Detection and Classification of Phytoplasma

Phytoplasmas methods and protocols described in the book edited by Musetti and Pagliari [88] are useful in laboratory practices for their detection and classification. Also, the technique illustrated by ISPM 27 [14] and Pusz-Bochenska et al. [89] for field- and laboratory-based assays of the pathogens and their insect vectors will be useful for less-equipped laboratories.

Currently, there are no curative control measures or identified resistant varieties. The control of vectors via synthetic insecticides and the eradication of infected plants are current management options [90]. Thus, early detection is critical for the removal of infected plants and strict quarantine measures to prevent the introduction and spread of the disease.

### 3.1. Detection of Phytoplasma

Initially, the identification and classification of Phytoplasmas were based primarily on biological properties such as symptoms, plant host range and relationships with insect vectors [91,92]. The symptomatology of Phytoplasma diseases is not sufficient as a diagnostic for identification and is not enough to distinguish among diverse Phytoplasma groups. Phytoplasmas cultured in axenic media and biochemical characterization reported by Contaldo and Bertaccini [83] and Contaldo et al. [40,41,42] represent the possibility of detection and the confirmation of Koch’s postulates. However, this prospect has not been widely accepted [16,36,43,93] preventing its use for the detection and taxonomic classification of Phytoplasmas. This should be reconsidered, as it will provide the pathogenicity, biochemical and morphological aspects of classifying the Phytoplasmas appropriately. In any case, there is always an open gap for future research in an attempt to further improve our understanding of the pathogen.

Other methods such as transmission electron microscopy (TEM), 4′,6-diamidino-2-phenylindole (DAPI) staining under fluorescence microscopy and enzyme-linked immunosorbent assay, have been developed for the detection of Phytoplasma diseases. These methods are laborious, time-consuming and unreliable. Molecular techniques using loop-mediated isothermal amplification (LAMP) and polymerase chain reaction (PCR) with restriction fragment length polymorphism analysis (RFLP) technology have aided in the detection and identification of diseases caused by Phytoplasmas [6,7,9,94,95,96,97,98].

The LAMP assay should be the method of choice for the early detection, diagnosis and monitoring of Phytoplasmas because of its suitability under field conditions. It requires minimal equipment, ease of use, minimal risk of sample contamination, less time for the whole process and visual confirmation of results [94,95,96,97,99,100]. Moreover, the LAMP amplification products can be confirmed by agarose gel electrophoresis if necessary. They have been used successfully with different nucleic acid extraction techniques for the detection of Napier grass stunt in Kenya and Ethiopia [94,95], coconut lethal yellow in Ghana and Mozambique [95,97], papaya dieback in Ethiopia [96,97] and grapevine yellow in South Africa [100]. However, there is a need to improve and simplify the nucleic acid extraction procedure.

Polymerase chain reaction assays such as nested and quantitative PCRs, microarrays and next-generation sequencing are now used for the detection of Phytoplasmas in both plants and insects [98]. It involves the sampling of tissues, extraction of DNA, selection of gene-specific primers that amplify a specific region of the 16S or 23S rDNA genes, PCR assays, RFLP or sequencing and sequence analysis [98].

Tissues should be selected from insect vectors and plant parts with phloem bundles, such as veins, mid-ribs and stalks, where the bacterium is most likely to be detected. A rapid and inexpensive crude sap nucleic acid extraction method for Phytoplasmas was reported by Minguzzi et al. [101]. An oligonucleotide that amplifies a specific region of the 16S or 23S rDNA genes, a spacer region between 16S and 23S, 23S, tuf, secA, secY, elongation factor EF-Tu and ribosomal proteins [4,102,103,104] are used to differentiate and identify Phytoplasmas (Table 3). In vitro and in silico RFLP analysis of Phytoplasma sequences from PCR-amplified rDNA provided a means to differentiate known and unknown Phytoplasmas into phylogenetic groups and subgroups [24,31,105].

**Table 3 microorganisms-13-01229-t003:** Primers used to detect plant infected by Phytoplasmas in Africa targeting the 16S rRNA.

Primers	Sequence (5′-3′)	Reference
P1	AAGAGTTTGATCCTGGCTCAGGATT	[106]
P4	GAAGTCTGCAACTCGACTTC	[107]
P6	CGGTAGGGATACCTTGTTACGACTTA	[106]
P7	CGTCCTTCATCGGCTCTT	[108]
R16F2n	GAAACGACTGCTAAGACTGG	[109]
R16R2	TGACGGGCGGTGTGTACAAACCCCG	[110]
LYDSR (Lethal Disease Tanzania)	GGTGCCATATATATTAGATTG	[111]
G813F (Lethal Disease Ghana)	CTAAGTGTCGGGGGTTTCC	[111]
AKSR (Lethal Disease Nigeria)	TTGAATAAGAGGAATGTGG	[111]
Rhode F (Lethal Disease Tanzania)	GAGTACTAAGTGTCGGGGCAA	[112]
Rhode R (Lethal Disease Tanzania)	AAAAACTCGCGTTTCAGCTAC	[112]

### 3.2. Classification System

Phytoplasmas are classified using 16S ribosomal RNA gene (rRNA) restriction fragment length polymorphism identity scores, whole-genome average nucleotide identity (ANI) or ecologically distinct host and molecular divergence [2,14].

Restriction fragment length polymorphism (RFLP) analysis of target gene differentiates sequences either into a “*Candidatus* Phytoplasma” genus based on percent sequence identity [2] or into ribosomal groups and subgroups on the basis of presence of restriction sites [31,113,114], with each group containing Roman numerals and subgroups designated by letters [115]. These classification systems, with the publication of Zhao et al. [116] resulted in the identification of 48 [33] or 49 [117] “*Candidatus* Phytoplasma” species in 37 groups (Table 1) and more than 150 subgroups.

The RFLP analysis of the 16S ribosomal RNA (rRNA) gene has been the most commonly used method for classification. Additionally, two web-based Phytoplasma classifiers namely iPhyClassifier (using rRNA sequences) [31] and CpnClassiPhyR (using the *cpn*60 gene sequence) [33], were developed for Phytoplasma classification.

Several limitations have been highlighted in the use of rRNA sequences alone [4,33,103,118,119,120] as a classification system for Phytoplasmas. These weaknesses have led to the design of classification schemes using several housekeeping genes from cpn60, rp, tuf, secY, etc. [4,119,120] and whole-genome sequence-based genotypic characterization [121]. These methods provide a better resolution of closely related taxa than 16Sr methods do [4,6,103,109,118,119,120,121,122]. The use of more than one gene for classification is referred to as multilocal sequence typing (MLST) analysis. This approach has been used for Phytoplasma subgroup differentiation and to modify earlier classifications and accurately identify new Phytoplasma strains [30,33,36,117,120,123,124].

In Africa, MLST was used by Zambon et al. [125] on grapevines from South Africa and Pilet et al. [36] on coconuts from Nigeria, Mozambique and Ghana to show the genetic diversity of “Ca. Phytoplasma asteris” and “Ca. P. palmicola”, respectively. Information on Phytoplasmas occurring in Africa using other genes is limited; thus, this review is based mainly on the RFLP rRNA classification system. This aims to provide information on reported Phytoplasmas in Africa for further study using MLST to validate their taxonomy. This suggests that MLST diagnostic studies will be highly useful to the phytosanitary community and other stakeholders within the region by providing appropriate markers for the surveillance and accurate reporting of Phytoplasmas occurring in the continent. Also, the development of a Phytoplasma classification web-based tool using several loci for the classification of Phytoplasma strains is suggested. More information can be explored on the proposed amendment of IRPCM [2] guideline on the description of Candidatus Phytoplasmas by Bertaccini et al. [117]. The clarifications and amendments required in the guidelines of Bertaccini et al. [117] were highlighted by [24].

## 4. Phytoplasma Diseases Status in Africa

Information on Phytoplasma diseases found in Africa were retrieved from a literature search, web sources (www.scholar.google.com) and GenBank, which were all accessed on or before 16th September 2024, as well as from the online global database of Hemiptera–Phytoplasma–Plant biological interactions (http://trivellone.speciesfile.org/) (accessed on 16 September 2024) [43]. In addition, information was derived from scientists working on Phytoplasma diseases through social media platforms and emails. The distribution of Phytoplasma diseases in Africa are indicated in Figure 1 and their relationships is shown in Figure 2. These Phytoplasmas belong to the 16SrI, 16SrII, 16SrIII, 16SrIV, 16SrVI, 16SrXI, 16SrXII, 16SrXIV and 16SrXXII groups.

### 4.1. Groups and Subgroups of Phytoplasmas in Africa

#### 4.1.1. Grapevine Yellows Disease

Grapevine yellows disease belongs to the aster yellows Phytoplasma group (16SrI). The disease was first reported in Africa on grapevines in 2004 in Tunisia and was delineated as 16SrI-B [64,126]. It was reported as a mixed Phytoplasma infection of 16SrXII-A and 16SrII-B in South Africa in 2006 [84] but was confirmed to be in the aster yellow subgroup of 16SrI-B [127,128,129]. Symptoms reported in both countries included veinal yellowing, necrosis, thicker and downward rolling leaves, shortened internodes, drooping, incomplete lignification and flexible shoots, abortion of growth tips and dry and shriveled immature bunches [84,126,128]. It is transmitted via vegetative planting materials and insect vectors [63,64]. The disease was detected in leafhoppers (*Mgenia fuscovaria*, *Aconurella prolixa*, *Cicadulina anestae*, *Austroagallia sinuata* and *Austrogallia cuneata*) [63,64] and planthopper (*Toya* sp.) [63] but are reported to be transmitted primarily by *M. fuscovaria* and possibly by *A. prolixa*. [63,72,73]. Alternative host plants of the disease were highlighted by Kruger et al. [63].

#### 4.1.2. Phyllody/Witches Broom/Virescence

Phytoplasmas belonging to group 16SrII are known to cause phyllody and witches’ broom disease in soyabean (*Glycine max*), eggplant (*Solanum melongena*), tomato (*Solanum lycopersicum*), squash (*Cucurbita pepo*), onion (*Allium cepa*), cactus (*Opuntia abjecta*), peanut (*Arachis hypogaea*), sunnhemp (*Crotalaria* sp.) and cotton (*Gossypium hirsutum*) in some African countries. Cotton phyllody reported on *Gossypium hirsutum*, *Sida cordifolia* and *Orosius cellulosus* in Burkina Faso [130,131,132] as well as *Gossypium hirsutum* and *Sida cordifolia* in Mali [118,133] were previously placed in the 16SrII-F subgroup [80] but were later reclassified as 16SrII-C [118]. The strains in Egypt affecting *Solanum melongena*, *Solanum lycopersicum*, *Cucurbita pepo*, *Allium cepa* and *Opuntia abjecta* were placed in the “Ca. P. Australasia” group and 16SrII-D subgroup [134,135,136]. Phyllody and witches’ broom in the 16SrII-C subgroup were reported on soybean in Mozambique and Malawi by Kumar et al. [137] and in Tanzania [138]. Alfaro-Fernández et al. [139] placed the causal Phytoplasma of Fababean phyllody (“Ca. P. aurantifolia”) in 16SrII-C subgroup, and it affects Faba bean (*Vicia faba*), Rattlepod (*Crotalaria saltiana*) and *Cicer arietinum* in Sudan.

The Phytoplasma that causes the phyllody disease of *Crotalaria saltiana* (“*Ca. P.* trifolii”) is in the 16SrVI-C group and was first reported in Sudan in 1962 on Periwinkle (*Catharanthus roseus*) [140]. The disease is associated with symptoms similar to those of Faba bean phyllody, with excessive proliferation of lateral shoots (witches’ broom) and small, chlorotic leaves with phyllody and virescence. Symptoms of cotton phyllody include shoot proliferations with shortened internodes, reduced leaflets and petioles while those for witches’ broom include phyllody, stunting, hairy root, abnormal colors on leaves and virescence [137,138]. *Orosius cellulosus* is an insect vector of this Phytoplasma.

Egyptian Phytoplasma virescence on Periwinkle (EF546439), which belongs to the aster yellows Phytoplasma group (16SrI), was reported by Omar et al. [141]. The diseased plant had few leaves, shortened internodes, virescence and witches’ broom symptoms. In Egypt, Gad et al. [142] reported on *Gazania rigens* (Gazania or Treasure flower) Phyllody Phytoplasma (MK 377249.1), which was associated with yellowing, proliferation, virescence and few leaves, as well as reduced flower size and stunted growth symptoms.

#### 4.1.3. Napier Grass Stunt Phytoplasma

Napier or Elephant grass (*Pennisetum purpureum*) is largely used as forage for cattle production in East Africa [143]. It is also used as biocontrol in a “push-pull” management system for the control of cereal stem borers (*Chilo partellus* and *Busseola fusca*) and fall armyworms (*Spodoptera frugiperda*) [144,145,146]. Napier grass stunt (NGS) disease caused by Phytoplasma is a serious disease of Napier grass resulting in 70–100% yield loss in infested farms [30,147]. The disease has been reported in Ethiopia, Kenya, Tanzania and Uganda [65,66,148,149,150].

On the basis of the 16S rDNA sequences, NGS in Ethiopia belongs to the 16Sr IIIA Phytoplasma group, a member of Candidatus Phytoplasma prunorum X-disease, which is closely related to the African sugarcane yellow leaf Phytoplasma (GenBank accession number AF056095) [65,66,148,149,150]. In Kenya, Tanzania and Uganda, “Ca. Phytoplasma oryzae” or rice yellow dwarf (RYD) Phytoplasma (GenBank accession number AY736374) is classified as a 16SrXI group member [27,148,149]. However, discussing the refinements of the 16SrXI and 16SrXIV groups using other genes, Abeysinghe et al. [30] believed that NGS should be reclassified as a new Ca. Phytoplasma species. To date, no reclassification of NGS has been performed, with 16SrXI retained by Fischer et al. [151] and Asudi et al. [152].

Infected NGS plants have small yellow leaves with the proliferation of tillers and shortening of internodes, a bushy appearance, pale yellow-green shoots and stunted growth, which results in low or no yield and ultimately, the death of the plants [148,149,153,154,155]. The disease is expressed in the regrowth of Napier grass after several cuttings or grazing by animals [27,155]. Spread occurs mainly through infected plant materials and insect vectors [27,147]. Vectors that have been reported include the leafhopper, *Maiestas banda* (Kramer) (Hemiptera: Cicadellidae) in Kenya and *Leptodel phaxdymas* and *Exiti anus* in Ethiopia [65,66,67]. The pathogen has been detected by Obura et al. [94] and Asudi et al. [27,154] in cereals, sugarcane and several asymptomatic wild grasses, which can serve as alternative hosts.

#### 4.1.4. Yellow Leaf Syndrome

Yellow leaf syndrome in sugarcane is associated with two pathogens: Phytoplasma (sugarcane yellow Phytoplasma) and virus (sugarcane yellow leaf luteovirus), which cause similar symptoms [156,157,158,159,160,161]. The International Society for Plant Pathology approved the use of “leaf yellow” and “yellow leaf” to distinguish the diseases caused by Phytoplasma and virus, respectively [162].

Sugarcane yellow leaf syndrome (SCYLS), referred to as “yellow wilt”, was first reported in Tanzania in the 1960s, with no specific pathogen listed [156,163,164]. The disease was attributed to Phytoplasmas and was found in Egypt, Kenya, Reunion, Senegal, South Africa, Swaziland, Uganda, Malawi, Mauritius, Morocco, Mozambique, Zambia and Zimbabwe [157,158,165,166]. The leaf yellow from South Africa was placed in the Western X (16SrIII) group (GenBank Accession No.AF056095) [30,157] and the 16SrI-B group (GenBank Acc. No. JX15763) from Egypt [167]. The characteristic symptoms are yellow discoloration along the midrib at the abaxial surface, sometimes with the lamina still green, shortening of terminal internodes, sucrose accumulation in midribs and necrosis of leaves starting from the leaf tips and then spreading through the leaf blade until the whole leaf is affected. In Africa, symptoms occur from the first three to five leaves, with visible dewlaps which often disappear with better growing conditions [166]. It is transmitted through infected planting materials and via insect vectors [168]. Its insect vector is yet to be identified in countries where leaf yellow has been reported in Africa.

#### 4.1.5. Sugarcane Grassy Shoot

Sugarcane grassy shoot is a member of the rice yellow dwarf Phytoplasma or 16SrXI group and is very closely related to the sugarcane white leaf, with a sequence similarity of more than 98% [30,169,170]. In Africa, it has been reported in Egypt (JN223446) [158] and Sudan [171]. It is transmitted by a yet-to-be identified insect vector(s) and through infected planting material in the two countries [158,171]. Infected plants produce numerous thin and slender tillers (with white or pale-yellow leaves) that give the plant a grassy or bushy appearance, and it does not produce any millable canes [30].

#### 4.1.6. Lethal Yellowing Diseases of Coconut and Cassava

Lethal yellowing diseases reported in Africa are classified into two Phytoplasma groups, 16SrIV and 16SrXXII, in different subgroups (Figure 1) and three *Candidatus* Phytoplasma spp., namely, Ca. P. cocostanizae, Ca. P. Palmicola and Ca. P. Palmae [28,30,36].

##### Lethal Yellowing Disease of Coconut

Lethal yellowing disease (LYD) has been reported in Nigeria, Benin, Togo, Ghana, Cameroon, Kenya, Tanzania and Mozambique [30,36,172,173,174,175]. Reports of Phytoplasma coconut disease date back to the early 1900s in Tanzania [36,176] and then in Nigeria in 1917 [38] and Kenya (Dowson, 1921 as cited by Pilet et al. [36]). In the 1930s, similar diseases were described in Togo, Cameroon and Ghana (Meiffren, 1951; Grimaldi and Monveiller, 1965; Chona and Addoh, 1970, all cited in Pilet et al. [36]). It was recorded in 1958 in Mozambique (de Carvalho and Mendes, 1958 cited by Eden-Green, [176]) and resulted in epidemics in the 1990s and 2014 in Mozambique and Cote d’Ivoire (Figure 3), respectively [36,177]. This disease is referred to as Awka wilt (Bronze leaf wilt) in Nigeria [38,172,178]; Cape St. Paul wilt in Ghana [179] and Cote d’Ivoire [177]; Kaıncope in Togo [180]; lethal disease in Tanzania; lethal yellowing in Mozambique and Kribi disease in Cameroun [181].

Ca. P. Palmicola has been reported in Ghana, Nigeria, the Ivory Coast and Mozambique [30,36,39,178,182], Ca. P. cocostanizae is present in Kenya and Tanzania (16SrIV-C) [183].

It has been reported in Refs. [28,184] and Mozambique (16SrIV-B and 16SrIV-C) [180,181,182] whereas Ca. P. palmae has not been reported in Africa [36,69]. Ca. P. Palmicola is classified into the 16SrXXIIA group, i.e., Ca. P. cocos nigeriae (Akwa wilt), reported in Nigeria and Mozambique, while Cape St. Paul wilt from Ghana and Cote d’Ivoire are classified into the 16SrXXIIB group [36,39,182,185].

Symptoms start with premature nut fall and necrosis (blackening) of inflorescences, followed by yellowing of the leaves (progressing from the older to younger leaves) and rotting of spear leaves; the whole leaves turn brown and break off leaving the trunk bare. This occurs within three to six months of initial symptoms [69,186].

It is transmitted through infected planting materials and insect vectors [36,69,185] and although the pathogen has been detected in the embryo [186,187], seed transmission has not been validated [36]. Myriads of insect vectors are suspected to be involved in the transmission of the pathogen in Mozambique, namely, *Diostrombusm kurangai*, *Meenoplus* sp. and *Platacantha lutea* [69]; *D. mkurangai* in Ghana; *Diostrombusm kurangai* [28,68]; *Meenoplus* sp. in Tanzania [28,68,188] and *Nedotepa curta* in Côte d’Ivoire [189], which are reported as potential vectors but have not been proven in successful transmission trials. In other affected African countries, no insect vector has been reported or suspected thus far. Lethal yellow has been reported on oil palm (*Elaeis guineensis*) and date palm (*Borassus aethiopium*) in Mozambique [184,190] and *Manihot* spp. in Cote d’Ivoire [191] as potential alternate hosts. Other suspected alternate hosts include plant species from the families Poaceae (*Paspalum vaginatum* Sw., *Pennisetum pedicillatum* Trin.), Verbenaceae (*Stachytarpheta indica* (L.) Vahl), Plantaginaceae (*Scoparia dulcis* L.), Phyllanthaceae (*Phyllantus muellerianus* (Kuntze) Excell), Cyperacea (*Diplacrum capitatum* (Willd.) Boeckeler), Fabaceae (*Desmodium adscendus* (Sw.) DC), Euphorbiaceae (*Manihot esculenta* Crantz), *Dalbergia saxatilis* and *Baphia nitida* [185,189,192,193].

##### Cassava Phytoplasma

In Africa, Phytoplasmas from the 16SrII and 16SrXXII groups have been reported in cassava plants. In Uganda, the restriction profiles obtained after RFLP of the PCR amplicons with the *Sau3* AI, *Hpa*II and *Hae*III enzymes were similar to those of “Ca. P. aurantifolia” (16SrII group). The 16S rRNA sequences of Phytoplasmas detected in cassava (EU315317) and four other nearby plant species (*Malvaviscus arborus* Cav (Malvaceae), *Codiaeum variegatum* (L.) A. Juss (Euphorbiaceae), *Hibiscus rosa-sinensis* L. (Malvaceae) and *Passiflora edulis* Sims (Passifloraceae) were 98% identical with that of the cactus witches’ broom Phytoplasma (AJ293216) [66,194]. Côte d’Ivoire cassava Phytoplasma of the 16SrXXII-B group, which was similar to coconut lethal yellow (CLYD) (16SrXXII-B), was recently identified in cassava from CLYD pandemic villages in Grand-Lahou, Côte d’Ivoire (KY563222) [191], indicating that it can attack another host crop (Figure 3c and Figure 4). Infected cassava plants exhibited leaf curling and yellowing in Côte d’Ivoire [191] and leaf yellowing, chlorosis, shortening of internodes and slight stunting in Kawanda, Uganda.

#### 4.1.7. Bermuda and Hyparrhenia Grass White Leaf

Bermuda grass white leaf belongs to the 16SrXIV Phytoplasmas or “*Candidatus* Phytoplasama cynodontis” group (GenBank Accession number AF100412). It has been detected in Egypt, Kenya, Sudan, Tanzania and Uganda. It was found in *Cynodon dactylon* and is characterized by stunted, bushy growth and small white leaves, shortened stolons or rhizomes, proliferation of axillary shoots and dead plants [27,150,158,195,196]. Hyparrhenia grass white leaf associated with a 16SrXI Phytoplasma has also been reported in Kenya [197].

#### 4.1.8. Phytoplasma Disease of Date Palm

White tip dieback (WDB) (GenBank Accession number AF100411) and Slow Decline (SD) (AF268000) are Phytoplasma diseases that affect mainly immature or 5–8 years old and mature date palm (*Phoenix dactylifera* L.), respectively, in Sudan [198,199]. Both of these strains belong to the 16SrXIV “*Candidatus* Phytoplasma cynodontis” group [198,199]. Severe chlorosis of the emerging spear leaf as well as tips of older frond leaflets are symptoms of WDB. The white chlorotic streaks with some necrosis extended longitudinally along the midrib, with the crown changing to dry white later. The plant dies within 6–12 months of symptom appearance [198]. Moreover, SD-infected palms die between 12 and 24 months after the first appearance of yellow leaves starting from the oldest frond and progressing to the young central fronds and spear leaves. These leaves turn white to light brown and break off, leaving the trunk bare [199]. Cronje et al. [198] also reported that young sprouts from infected SD have yellow fronds and as the crown dies, the spear leaf becomes whitish, necrotic and easily removed, resulting in rot-smelling basal tissues. The first report [200] confirmed the presence of streak yellows on date palm in Egypt called Al-Wijam disease (Abou-El-Einin, 2010) which was described as a member of the aster yellow Phytoplasma 16SrI group (KF826615) [201]. The most common symptom is longitudinal streak yellowing along the midrib and premature drying of young and mature leaflets as well as stunted growth.

#### 4.1.9. Phytoplasma Diseases of Papaya

Phytoplasma diseases of papaya, such as papaya dieback, yellow crinkle, mottle, mosaic and papaya bunchy top, have been reported worldwide [66,202,203,204,205,206]. In Africa, papaya dieback (PDB), classified as 16SrII, has been reported in Ethiopia (Acc. No. DQ285659) [206], papaya bunchy top (PBT) in Nigeria (Figure 5a) is designated as subgroup 16SrXII-O in the Stolbur Phytoplasma group (MW530522; MW530532; MW530524) [207] and unnamed in Cote d’Ivoire in the aster yellow subgroup of 16SrIB (PPB820865, PPB820866, PPB820867) [208] (Figure 5b,c). The symptom expression for PDB is a bright yellowing of the upper young leaves, followed by mosaic, crinkling, leaf tip necrosis, and drying of the upper leaves, leading to death of the infected plants [206]. The symptoms observed for PBT in Nigeria include leaf yellowing and crinkling, bending of the petioles and shoot at an angle, premature fruit drop and rot, necrosis of the leaf veins and leaf margins, axillary shoot proliferation in the apical crown or near the top of the plant and dieback of the entire plant or side shooting at the lower stem region [207]. This Phytoplasma has also been found in tomato (OP123558) and jute mallow (OP123559) in Nigeria [209]. Symptoms observed in Phytoplasma that affected papaya in Cote d’Ivoire are leaf edge curling and mosaic [208].

#### 4.1.10. Unclassified Phytoplasma Group

Goosegrass white leaf of unknown 16Sr group has been detected in Kenya, Tanzania and Uganda [27,150,196]. In Egypt, on the basis of electron microscopy results, a Phytoplasma associated with the malformation of mango fruits [210] and carrot plants (*Dacus carota*) transmitted by *Hebata decipiens* [74] has been reported but not yet fully characterized [74,135].

## 5. Impact of Phytoplasma Diseases in Africa

With improved molecular diagnostic techniques, the list of diseases caused by Phytoplasmas continues to increase because those diseases previously attributed to other pathogens or of unknown etiology are being identified and properly classified. The situation is further exacerbated by the fact that there are no viable management options when a plant is infected, and only a few vectors of Phytoplasmas diseases have been identified, thus limiting preventive/control options. The Phytoplasma disease pandemic experienced by some countries has brought the socio-economic importance of the disease to the fore. In Africa, only a few reports outlining their socio-economic effects exist, mainly due to inadequate studies of their epidemiology, awareness of the disease and their non-culturable nature.

The wine industry in South Africa produces approximately 3.9% of the world’s wine (ranked 7th), contributing ZAR 56.5 billion to the GDP [211]. Approximately 87,848 ha is cultivated, with white varieties constituting 55% of the vines planted [211]. The presence of aster yellows disease threatens this industry [128,212] by reducing the quality of grapevines, and lowering the yield to death of the vines, leading to financial losses for vineyard owners and the broader economy [212,213]. Collaborations between the research institute and viticulture industry have attempted to curtail this disease, but it remains an important quarantine pest, with delineated surveys being conducted regularly to restrict spread [72,73,212]. An epidemiology study by Carstens [212] indicated that Chenin Blanc, Chardonnay and Pinotage, which belong to some of the ten most cultivated varieties [211], have a high incidence of the disease.

Coconut lethal yellow disease (CLYD) outbreaks in some African countries resulted in the loss of millions of coconut palms [175,192,214]. The international trade of coconut seedlings has been drastically affected due to phytosanitary concerns. This has had a negative impact on the cultivation, marketing and processing of coconut as well as the diverse uses of different parts of the palm tree as sources of income, staple foods and livelihoods in affected countries [193]. Mozambique lost its ranking as the top producer of coconut in Africa to Nigeria due to CLYD after the first and second reported incidences in 1992 and 2010, respectively [190].

*Pennisetum purpureum*, a major fodder crop for dairy cattle in Eastern Africa, has experienced a 70–100% reduction in growth as a result of Napier grass stunt [10,65,149,153,155,215]. This has resulted in reduced livestock rearing due to scarce fodder [153,155,215,216].

## 6. Conclusions and Prospects

With increasing new outbreak and reports of previously uncharacterized Phytoplasma diseases worldwide [3,5,11,69,82,126,128,137,138,149,177,191,208,217,218,219], Africa need to be alert and well prepared. Multiplicity of Phytoplasmas affecting different plants in Africa has remained undetected and unreported mainly due to inadequate awareness of the pathogens and lack of molecular detection facilities in the tropics. Survey and diagnosis of the leafhopper (Auchenorrhyncha: Cicadellidae), planthopper (Auchenorrhyncha: Fulgoromorpha) and psyllid (Sternorrhyncha: Psyllidae) populations in African countries might reveal possible unknown Phytoplasma species. Identification of Phytoplasma and host species would improve our understanding of their epidemiology and economic impacts and contribute to the development of management strategies that prevent escalation into outbreaks and major disease such as the Napier grass stunt Phytoplasma and Coconut lethal yellow diseases. Hence, further investigations are needed to identify and/or develop Phytoplasma detection techniques appropriate for less developed laboratories in Africa. This will also enhance the identification of other potential insect vectors and hosts.

## Figures and Tables

**Figure 1 microorganisms-13-01229-f001:**
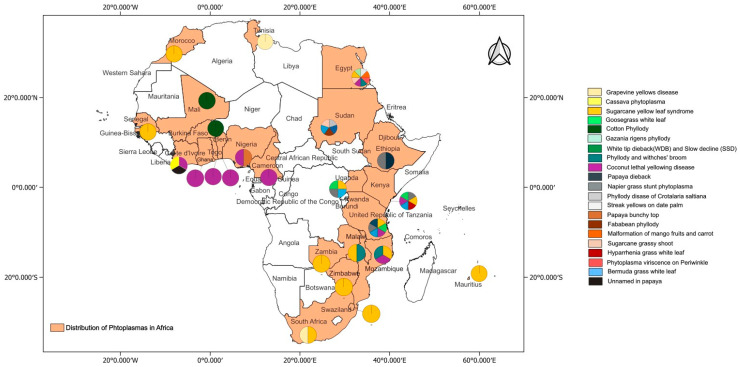
Phytoplasma diseases reported in countries in Africa.

**Figure 2 microorganisms-13-01229-f002:**
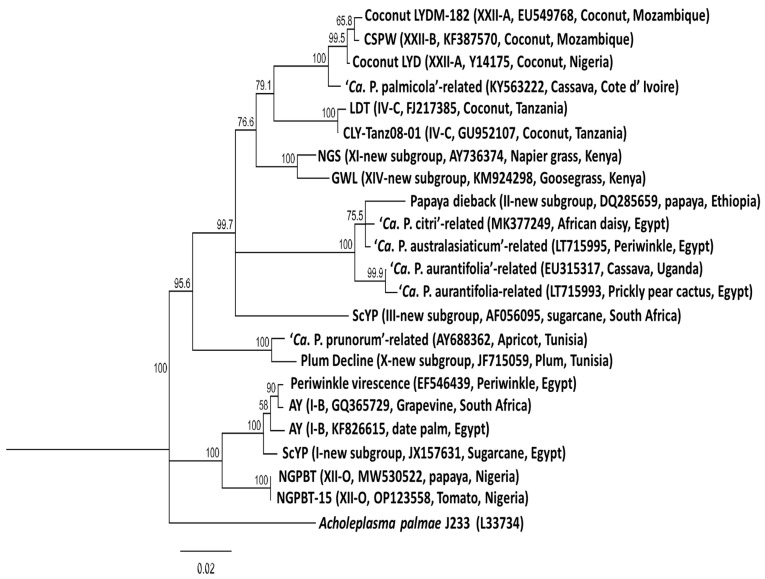
Phylogenetic tree constructed from published 16S rRNA gene sequences of the Phytoplasma species reported in Africa employing the neighboring-joining method using the MEGA software, version 7. The percentage of replicate trees in which the associated taxa clustered together in the bootstrap test (1000) is shown next to the branches. Branch lengths are proportional to the number of inferred character state change. The scale bar length represents the number of nucleotide substitutions per site. The 16S rRNA sequence of *Acholeplasma palmae* (L33734) was used as an outgroup.

**Figure 3 microorganisms-13-01229-f003:**
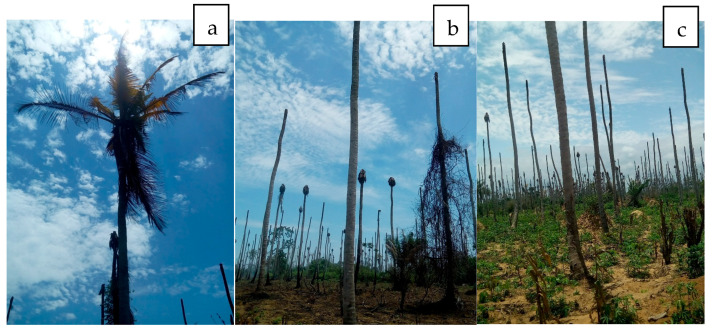
(**a**,**b**) Coconut lethal yellow disease destruction of coconut plantations in Braffedon, Grand Lahou, Cote d’Ivoire. (**c**) Land is now being used for Cassava cultivation.

**Figure 4 microorganisms-13-01229-f004:**
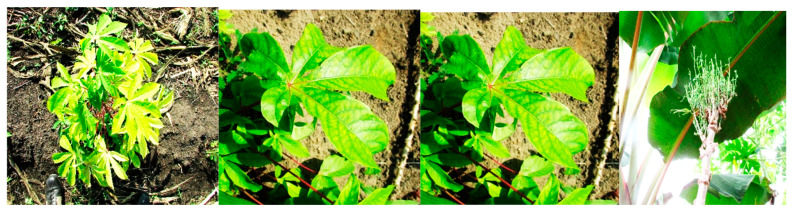
Cassava Phytoplasma disease in Côte d’Ivoire. Source: Plant Health Unit, University Nangui Abrogoua, Côte d’Ivoire.

**Figure 5 microorganisms-13-01229-f005:**
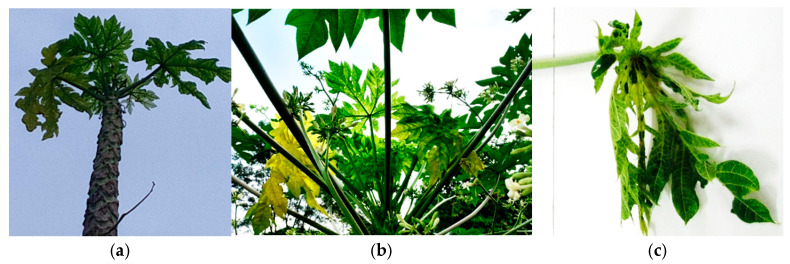
Phytoplasma diseases of papaya in (**a**) Nigeria and (**b**,**c**) Cote d’Ivoire. Source: Plant Health Unit UNA-CI.

## Data Availability

The original contributions presented in this study are included in the article. Further inquiries can be directed to the corresponding authors.

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
