# Peer review of "Status and Distribution of Diseases Caused by Phytoplasmas in Africa"

_microorganisms, 2025, doi:10.3390/microorganisms13061229_

Round 1

Reviewer 1 Report

Comments and Suggestions for Authors The manuscript "Status and distribution of diseases caused by phytoplasmas in Africa" by Shakiru Adewale Kazeem et al. presents a report on widespread phytoplasmoses in Africa and their current status. The article as a whole is systematized and valuable, but there are a number of comments on it.     1. The authors mention the prevalence of phytoplasma diseases in Africa, but there is no detailed information on individual countries and crops. It seems to me that in such works to systematize a large amount of information, it is necessary to collect and provide information in a table, which would increase the interest of readers. 2. Although economic damage is mentioned in the work, there is no quantification of losses from diseases, for example, loss of crop yields or farmers' incomes in individual countries. 3. Unfortunately, the work is limited to recommending early detection and destruction of infected plants, but other control measures, including integrated approaches, have been developed to control phytoplasmas. 4. There are no details of LAMP and PCR systems in the work, there is no detailed description of the protocols, including the reagents used, concentrations and conditions of analysis. 5. The paper notes the role of alternative hosts, but there is no list of wild plants in Africa that may be reservoirs of phytoplasmas. 6. The authors discuss possible insect vectors. However, it would be good to present this in the form of a table or a systematic list of confirmed and suspected vector species by country and culture. 7. The authors note the need for further research, but do not formulate clear directions: seasonality, environmental factors, behavior of vectors, etc. This needs to be fixed. 8. The authors mention the role of seeds in the possible transmission of phytoplasmas, but there is no risk analysis or recommendations for quarantine services. 9. Despite the fact that the overview is presented for Africa, it would be good to present a comparison with other regions of the world.  10. Unfortunately, the issue of the potential interaction of phytoplasmas with other pathogens in the form of mixed infections has not been worked out. 11. Some references in the literature list date back to the early 2000s, which needs to be updated with new works. 12. It would be cool and clear to provide a map visualizing the distribution zones of phytoplasmas in Africa. 13. Despite the impressive amount of work, it is not discussed how global climate change may affect the spread of phytoplasma diseases and vectors in Africa. 14. The paper analyzes the main crops (grapes, coconut), but there is no information on important grain or vegetable crops.   Correcting these inaccuracies would improve the quality of the article.

Author Response

Reviewer 1

The manuscript "Status and distribution of diseases caused by phytoplasmas in Africa" by Shakiru Adewale Kazeem et al. presents a report on widespread phytoplasmoses in Africa and their current status. The article as a whole is systematized and valuable, but there are a number of comments on it.    

Thank you very much for finding the time to rigorously going through our manuscript and the constructive suggestions. Our responses are indicated under each comment.

1. The authors mention the prevalence of phytoplasma diseases in Africa, but there is no detailed information on individual countries and crops. It seems to me that in such works to systematize a large amount of information, it is necessary to collect and provide information in a table, which would increase the interest of readers.

Response:  Under the session “Phytoplasma diseases status in Africa”, this has been addressed. Figure 1, the phylogenic tree has included this information. Also, the information except the crops have been provided in African maps that was recommended by the reviewer.

  1. Although economic damage is mentioned in the work, there is no quantification of losses from diseases, for example, loss of crop yields or farmers' incomes in individual countries.

Response:  Phytoplasma diseases research is still in the infancy in Africa, information on economic impact on crops/farmers/countries is scarce. The few we could find are in Session 5.0 Impact of phytoplasmal disease in Africa.

  1. Unfortunately, the work is limited to recommending early detection and destruction of infected plants, but other control measures, including integrated approaches, have been developed to control phytoplasmas.

Response:  As mentioned in Section 3 “Detection and classification of phytoplasma”. There is limited recommendation for control of phytoplasma disease. Yes, IPM is feasible when you have methods can that work singly or together which is not the case with phytoplasma. Resistant varieties are limited, no curative measure except reports of antibiotics use. In other words, IPM strategy for phytoplasma control information is limited.

  1. There are no details of LAMP and PCR systems in the work, there is no detailed description of the protocols, including the reagents used, concentrations and conditions of analysis.

Response:  We have provided the references for these techniques and listed the Primers used in Table 2 and discuss the benefit for each.

  1. The paper notes the role of alternative hosts, but there is no list of wild plants in Africa that may be reservoirs of phytoplasmas.

Response: There is dearth of information on this aspect too. The ones we could find were listed against each reported Phytoplasmal disease in Africa with references provided.

  1. The authors discuss possible insect vectors. However, it would be good to present this in the form of a table or a systematic list of confirmed and suspected vector species by country and culture.

Response:  Thank you. We have added it to the manuscripts

  1. The authors note the need for further research, but do not formulate clear directions: seasonality, environmental factors, behavior of vectors, etc. This needs to be fixed.

Response:  Thank you very much. We have in the Conclusion and Prospect.

  1. The authors mention the role of seeds in the possible transmission of phytoplasmas, but there is no risk analysis or recommendations for quarantine services.

Response:  Thank you very much. We have in section 2.2.2

  1. Despite the fact that the overview is presented for Africa, it would be good to present a comparison with other regions of the world. 

Response:  Thank you for the suggestion. The first three sessions carried this information. If we start to mention the distribution or impact where the disease occurred in other countries for example, the information on Africa will be lost, since not much have been done in Africa.

  1. Unfortunately, the issue of the potential interaction of phytoplasmas with other pathogens in the form of mixed infections has not been worked out.

Response:  Thank you. We agree but some works have proved that previous report were actually caused by phytoplasma. In some cases, with mix infection. It was mentioned.

  1. Some references in the literature list date back to the early 2000s, which needs to be updated with new works.

Response:  Yes, because some were first report of the disease, and no subsequent work has been done on them. In some cases, we equally cited recent works where they were mentioned. Equally, there is limited information on phytoplasma research in Africa

  1. It would be cool and clear to provide a map visualizing the distribution zones of phytoplasmas in Africa.

Response:  Thank you very much. We have added it to the manuscripts

  1. Despite the impressive amount of work, it is not discussed how global climate change may affect the spread of phytoplasma diseases and vectors in Africa.

Response: We have limited references on this also especially for Africa

  1. The paper analyzes the main crops (grapes, coconut), but there is no information on important grain or vegetable crops.   Correcting these inaccuracies would improve the quality of the article.

Response:  We don’t have much information on reported cases on grains/vegetable crops mentioned in Africa

Reviewer 2 Report

Comments and Suggestions for Authors

The paper “Status and distribution of diseases caused by Phytoplasmas in Africa” concerns a complex topic in relation to an area of ​​the planet where information is still limited. The paper is well written, with a clear general organization. The bibliographical sources are up to date and sufficient. I think this review is important and I have a few observations to make.

L35, L41, L56 and others sentences in the whole paper: please be consistent with capital letters (eg. phytoplasmas). Use it when you refer to the genera, not as common name.

L44: “Lethal yellow disease” is a very generic term which, to my knowledge, is not related to a specific disease. You're probably referring to coconut, so I suggest being more specific.

Table 1: to my knowledge, some important strains missing: Ca. P. australiense, Ca. P. Australasia, Buckland Valley grapevine  yellows phytoplasma, Ca. P. omanense. Furthermore, probably the most important phytoplasma, the Grapevine flavescence dorée phytoplasma (ex Ca. P. vitis) is missing.

L68: please consider recent changes in taxonomy, as not all phytoplasmas fit into the Candidatus scheme (eg. Grapevine flavescence dorée phytoplasma, no more a Candidatus one)

General comment: Does chapter 2 and 3 refer to general aspects or only to those relating to phytoplasmas found in Africa? I have the impression that the general aspects are considered, for which several reviews are already available. Furthermore, what the authors indicate is not sufficiently in-depth for a general review. In my opinion it is necessary to eliminate chapter 2.1 relating to symptoms (the specific one for the various diseases in Africa is already present). Chapter 3 should be even more specific to diseases present in Africa and, therefore, follow Chapter 4.

Table 2: many important phytoplasma can be directly detected by real time PCR. Generally speaking, this review seems not to consider these aspects which, instead, should be reported.

L266-271: as reported before, some phytoplasma no more belong to Candidatus.

L313: please rephrase in relation to Africa, otherwise it seems that Flavescence doree or Bois noir, the main yellows of grapevine, do not exist.

L550: in other parts it was reported as LYD, but I think that CLYD is more appropriate.

General comment: a figure/map showing the distribution of the main phytoplasmas in the different African nations is essential.

Author Response

The paper “Status and distribution of diseases caused by Phytoplasmas in Africa” concerns a complex topic in relation to an area of ​​the planet where information is still limited. The paper is well written, with a clear general organization. The bibliographical sources are up to date and sufficient. I think this review is important and I have a few observations to make.

Thank you very much for finding the time to rigorously going through our manuscript and the constructive suggestions. Our responses are indicated under each comment.

L35, L41, L56 and others sentences in the whole paper: please be consistent with capital letters (eg. phytoplasmas). Use it when you refer to the genera, not as common name.

Response: Thank you very much. It has been corrected

L44: “Lethal yellow disease” is a very generic term which, to my knowledge, is not related to a specific disease. You're probably referring to coconut, so I suggest being more specific.

Response:  Lethal Yellow was meant to cover that for coconut and cassava. Lethal Yellow was also reported in cassava.

Table 1: to my knowledge, some important strains missing: Ca. P. australiense, Ca. P. Australasia, Buckland Valley grapevine yellows phytoplasma, Ca. P. omanense. Furthermore, probably the most important phytoplasma, the Grapevine flavescence dorée phytoplasma (ex Ca. P. vitis) is missing.

Response: Thank you for the observation. It has been corrected. Please go through the sources of the Table 1: Some of these strains mentioned have been abolished, which is why they were not listed on the Table

L68: please consider recent changes in taxonomy, as not all phytoplasmas fit into the Candidatus scheme (eg. Grapevine flavescence dorée phytoplasma, no more a Candidatus one)

Response: Grapevine yellow disease was mentioned in the manuscript and is still classified in the Candidatus scheme, it is different from Grapevine flavescence dorée phytoplasma and Bois noir which have not been reported in Africa.

General comment: Does chapter 2 and 3 refer to general aspects or only to those relating to phytoplasmas found in Africa? I have the impression that the general aspects are considered, for which several reviews are already available. Furthermore, what the authors indicate is not sufficiently in-depth for a general review. In my opinion it is necessary to eliminate chapter 2.1 relating to symptoms (the specific one for the various diseases in Africa is already present). Chapter 3 should be even more specific to diseases present in Africa and, therefore, follow Chapter 4.

Response:   We understand but we need to present theses information to readers who are not familiar with the disease especially in Africa.

Table 2: many important phytoplasma can be directly detected by real time PCR. Generally speaking, this review seems not to consider these aspects which, instead, should be reported.

Response: This was mentioned in the Section 3.1 Detection of phytoplasma as quantitative PCR

L266-271: as reported before, some phytoplasma no more belong to Candidatus.

Response:  We have responded above. It is still classified in the Candidatus scheme. Please see Table 1 sources.

L313: please rephrase in relation to Africa, otherwise it seems that Flavescence doree or Bois noir, the main yellows of grapevine, do not exist.

Response: Grapevine yellows phytoplasma was used throughout the manuscript.

L550: in other parts it was reported as LYD, but I think that CLYD is more appropriate.

Response: we are using it the way it was reported and based on the crop attacked.

General comment: a figure/map showing the distribution of the main phytoplasmas in the different African nations is essential.

Response:  Thank you very much. We have added it to the manuscripts

Round 2

Reviewer 1 Report

Comments and Suggestions for Authors

The authors have corrected the text on most of the comments, but a number of others have appeared. For example, the names of the drawings are not completely complete. In table 1, we should also add a column on the plants on which they are found. In table 2, the gene with which the primers interact should be indicated and these primers should be divided according to the species with which they interact. Figure 1 shows not phytoplasmas, but a phylogenetic tree based on phytoplasma isolates isolated in Africa. In addition, it is not specified what the scale means and was used as an outgroup, in which program the tree was made.

Author Response

Once again thank you for your time and constructive comments. We have addressed them. Please find below.

The authors have corrected the text on most of the comments, but a number of others have appeared. For example, the names of the drawings are not completely complete.

In table 1, we should also add a column on the plants on which they are found.

Comments: It has been added

In table 2, the gene with which the primers interact should be indicated and these primers should be divided according to the species with which they interact.

Comments: This is referring to Table 3? It has been corrected with the targeted gene indicated. Some of them are universal primers while others are not which is why the references are provided.

Figure 1 shows not phytoplasmas, but a phylogenetic tree based on phytoplasma isolates isolated in Africa. In addition, it is not specified what the scale means and was used as an outgroup, in which program the tree was made.

Comments: Corrected